# Neural Logical Index for Fast Knowledge Graph Complex Query Answering

## Abstract

Complex Query Answering (CQA) over knowledge graphs is a crucial multi-hop reasoning task aimed at addressing first-order logical queries within large and incomplete knowledge graphs. Direct traversal search methods rely solely on graph topology and often miss answers due to the incompleteness of the graph, thus neural models have been proposed to generalize the neglected answers from observed facts. There are primarily two lines of research tackling the challenges of CQA. Query embedding models learn representations for complex queries, offering fast speed but often providing only generic performance. In contrast, neural symbolic search methods deliver better performance, although they tend to be computationally more expensive. In this paper, we propose an efficient and scalable search framework that combines the precision of symbolic methods with the speed of embedding techniques. Our model utilizes embedding methods to compute Neural Logical Indices (NLI) to reduce the search domain for each variable in advance, followed by an approximate symbolic search for fine ranking. The search is precise for tree-form queries and approximates cyclic queries (which are NP-complete) in quadratic complexity with respect to the search domain, matching the complexity of tree-form queries. Experiments on various CQA benchmarks show that our framework reduces computation by 90% with a minimal performance loss, alleviating both efficiency and scalability issues for symbolic search methods. Our code is provided in `https://anonymous.4open.science/r/efficient_CQA/README.md`.

## 1 Introduction

Knowledge Graphs (KGs) are knowledge bases that represent relational facts in graph form. Although KGs have an interpretable structure supporting many real-world applications (Ji et al., 2021), they often suffer from incompleteness (Safavi & Koutra, 2020; Hu et al., 2020). Recently, complex query answering (CQA) (Ren et al., 2023; Wang et al., 2022) over knowledge graphs has attracted significant interest because this practical task performs logical reasoning with new knowledge inferred from observed knowledge graphs. Currently, the CQA task mainly focuses on answering existential first-order logic queries (Ren & Leskovec, 2020; Yin et al., 2023), involving logical operations such as conjunction, negation, disjunction, and the existential quantifier. Due to the incompleteness of KGs, many answers are overlooked in direct traversal searching.

There are primarily two lines of research to address the challenge of CQA. One is query embedding methods which represent the query's answer set using representations like vector, box, beta distribution of low dimensional space (Hamilton et al., 2018; Ren et al., 2020; Ren & Leskovec, 2020). In this approach, logical operations are transformed into set operations within an operator tree (Wang et al., 2021; Ren et al., 2023), modeled by neural networks in alignment with their semantics in the low-dimensional space. Although the representational capabilities have been thoroughly explored (Zhang et al., 2021; Choudhary et al., 2021), current query embedding methods still face limitations in both performance and expressiveness (Yin et al., 2024). The second line of research, neural-symbolic search methods (Arakelyan et al., 2020; Zhu et al., 2022; Bai et al., 2023; Yin et al., 2024), utilizes knowledge graph completion methods (Bordes et al., 2013; Sun et al., 2018; **?**) as a backbone to predict missing facts and model logical operations using fuzzy logic inference. Though symbolic search methods usually have both strong performance and interpretability, they typically suffer from high complexity and lack scalability, as shown in Fig 1.

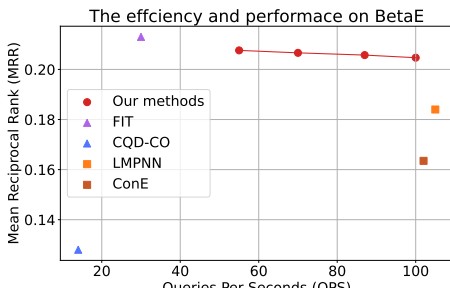 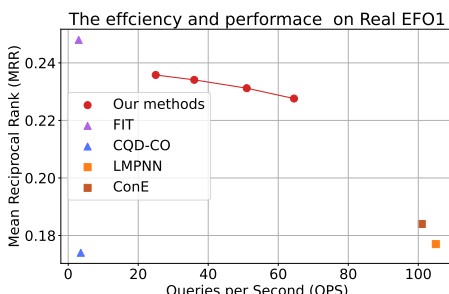

Figure 1: The performance and efficiency on the BetaE (Ren & Leskovec, 2020) and Real EFO1 (Yin et al., 2024) benchmarks. We use Mean Reciprocal Rank (MRR) and Queries Per Second (QPS) as metrics for performance and efficiency, with **higher values indicating better results**. Our proposed framework flexibly reduces the search domain for symbolic methods, and strikes a good balance between performance and efficiency. We present the results of the search domain as 1000, 2000, 3000, and 4000 for the FB15K-237 knowledge graph with 14,951 entities. With a reduced search domain, our method can achieve significant improvements in efficiency with a slight decrease in performance.

The differences between the embedding-based approach and the neural-symbolic search approach are not just empirical but fundamentally related to query syntax and complexity (Yin et al., 2024). From a syntactical aspect, the widely used operator tree in query embedding methods is limited to representing a subset of Existential First Order queries with single free variable (EFO1 queries), denoted as tree-form queries (Yin et al., 2024). The tree-form queries in previous datasets (Ren et al., 2020; Ren & Leskovec, 2020; Wang et al., 2021) are solvable with quadratic complexity concerning $|\mathcal{E}|$, where $|\mathcal{E}|$ is denoted as the number of entities in KG (Bai et al., 2023; Yin et al., 2024). Consequently, the real EFO1 dataset (Yin et al., 2024) that includes multigraphs and cyclic graph patterns has been proposed, further underscoring the complexity issues. When answering the EFO1 query with $n$ variable, the existing precise symbolic search method (Yin et al., 2024) exhibits a worst-case computational complexity $O(|\mathcal{E}|^n)$ which grows polynomially with the size of the knowledge graph but increases exponentially with the number of variables. The query embedding methods can only approximate the general EFO1 query by operator tree but the complexity is polynomial with respect to its dimension.

Despite the fundamental differences between these two types of methods, we design a synergistic way to integrate them together in a mutually beneficial way, thereby delivering a new frontier of performance and efficiency, as shown in Fig 1. Inspired from the arc consistency (Chen et al., 2011) in constraint satisfaction problem (Gottlob et al., 2000; Tönshoff et al., 2022)[1], it is unnecessary to use all entities of the KG as the search domain for symbolic search methods. Instead, we can leverage the surrounding constraints of each variable to reduce the corresponding search domain in advance, thereby decreasing the computational cost required for originally slow but accurate symbolic searches. Our first contribution is the Neural Logical Index (NLI), which models the surrounding constraints with fast but not accurate embedding-based methods. In terms of the extent of the constraints utilized, we propose two specific strategies, as illustrated in Fig. 2. The **local constraints** strategy uses the relations directly connected with the variable, with a relation tail prediction task formulated to assist in computing the local constraints. The **global constraints** strategy considers broader constraints across the entire query to further reduce the search domain. Further details are presented in Section 3.

The second technical contribution of this work is that we propose a scalable framework, Neural Logical Index for Search Approximately (NLISA) addressing the cyclic query in quadratic complexity with respect to the search domain. In particular, this approximate search framework can be parallelized and is exact in tree-form queries similar to QTO (Bai et al., 2023) and FIT (Yin et al., 2024). Combining the approximate search framework with neural logical indices, our method can efficiently answer the general EFO1 queries, including the cyclic queries. Experiments on various CQA benchmarks show that our framework reduces computation by 90% with a minimal performance loss, alleviating both efficiency and scalability issues. Additionally, we demonstrate that our framework can execute neural-symbolic methods on a KG with an order of magnitude more entities than before, highlighting the scalable nature of our approach.

---

[1]The complex query answering can be reduced as the constraint satisfaction problem by treating each atom in the logical query as constraints.

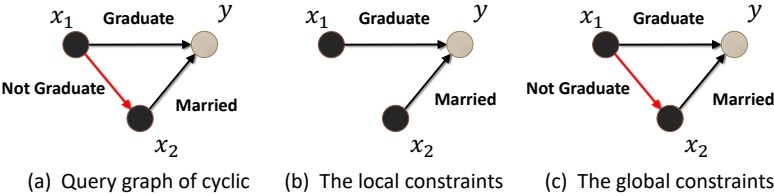

Figure 2: Left is the query graph of a given query "Find someone who is married to a person who graduated from a different institution." The formal language is $\exists x_1, x_2. \neg Graduate(x_1, x_2) \land Graduate(x_1, y) \land Married(x_2, y)$. This cyclic query can not be modeled by an operator tree. The middle is the presentation of the used constraints in the local strategy for free variable $y$. The right is the presentation of the used constraints in the global strategy for the free variable $y$.

## 2 BACKGROUND

### 2.1 KNOWLEDGE GRAPH

**Definition 1** (Knowledge Graph). *Let $\mathcal{E}$ be the finite set of entities and $\mathcal{R}$ be the finite set of relations, a knowledge graph is a collection of factual triples $\mathcal{G} = \{(s_i, r_i, o_i)\}$, where $s_i$ and $o_i$ are entity objects, and $r_i$ is a relation predicate.*

We augment the facts by adding reverse relations and denote $r_+$ as the original relation $r$ and $r_-$ as its reverse, where each original triple $(h, r, t)$ will result in two triples: $(h, r_+, t)$ and $(t, r_-, h)$. The knowledge graph can be represented as a first-order logic knowledge base, where each triple $(s, r, o)$ denotes an atomic formula $r(s, o)$, with $r \in \mathcal{R}$ a binary predicate and $s, o \in \mathcal{E}$ its arguments.

### 2.2 LOGICAL QUERIES AND ANSWER SET

Complex Query Answering (CQA) on knowledge graphs aims to derive the answer set of the multi-hop logical query using the KG as the knowledge base. The existential first order logic queries with single free variable (EFO1) involving the existential quantifiers ($\exists$), conjunction ($\land$), disjunction ($\lor$), and negation ($\neg$), are of particular interest in Disjunctive Normal Form (DNF).

**Definition 2** (EFO1 Query). *The EFO1 query is defined as :*

$$\psi(y; x_1, \cdots, x_n) = \exists x_1, \cdots x_n. (c_1^1 \land \cdots \land c_{n_1}^1) \lor \cdots \lor (c_1^k \land \cdots \land c_{n_k}^k), \quad (1)$$

*where $c_j^i$ is the atomic formula $r(h, t)$ or its negation $\neg r(h, t)$, $r$ is a relation predict from $\mathcal{R}$, $h$ and $t$ are entity belong to $\mathcal{E}$ or a variable ranging from $\mathcal{E}$.*

To simplify notation, we sometimes denote $\psi(y; x_1, \cdots, x_n)$ as $\psi(y)$ or $\psi$. Given a variable $x$ or $y$ in query $\psi$, and any entity $s \in \mathcal{E}$, the substitution involves replacing all occurrences of the variable in $\psi$ with $s$, denote the process as $s/x$ or $s/y$. Then we can define the answer set of the logical query.

**Definition 3** (Answer set). *Given an EFO1 query $\psi(y)$, the answer set is defined by*

$$\mathcal{A}[\psi(y)] = \{s \in \mathcal{E} | \psi(s/y) = \text{TRUE}\}. \quad (2)$$

By the DNF, the answer set of EFO1 query $\psi$ can be the union of decomposed conjunctive query's answer set (Ren et al., 2020), and we denote $\phi$ for conjunctive query. The conjunctive query can be reduced as the constraint satisfaction problem by treating each atom as constraints (Yin et al., 2024).

### 2.3 OPERATOR TREE AND QUERY GRAPHS

By replacing logical operations with their corresponding set operations, some logical queries can be represented in the operator tree (Ren et al., 2020). Notably, the existential quantifier introduces a new set operation, set projection, which corresponds to logic skolemization (Luus et al., 2021).

As illustrated in Fig. 2, cyclic queries cannot be modeled using operator trees; thus, we follow Yin et al. (2023) in using query graphs to represent general EFO1 queries. Each conjunctive query $\phi$ is represented as a query graph, then the DNF can be represented as the disjoint query graphs.

**Definition 4** (Query Graph). *Let $\phi$ be a conjunctive query. Its query graph $G_\phi = \{(h_i, r_i, t_i, \text{NEG}_i)\}$ consists of quadruples, where each quadruple corresponds to an atomic formula or its negation. This representation defines an edge with two endpoints $h$ and $t$, along with two attributes: $r$, which denotes the relation, and $\text{NEG}_i$, which is the bool variable indicating whether the atom is positive.*

**Definition 5** (Neighbor Subgraph). *Let $G_\phi$ be a conjunctive query graph and $x$ be the variable in $G_\phi$, the edges of neighbor subgraph for $x_i$ is $\mathcal{N}_e(x_i, G_\phi) = \{(h_i, r_i, t_i, \text{NEG}_i) \in G_\phi | h_i = x \text{ or } t_i = x\}$, formed from the neighbor constraints of $x_i$. The $\mathcal{N}_n(x_i, G_\phi)$ is the corresponding node set.*

We present an example of a query graph in Fig. 2. The concept of query graph is also similar to the constraint graph (Vardi, 2000), as explored in constraint programming problems.

## 2.4 KNOWLEDGE GRAPH COMPLETION AND KNOWLEDGE GRAPH EMBEDDING

The task of knowledge graph completion addresses the issue of missing edges in a knowledge graph by predicting the tail entity given the head entity and relation $(s, r, ?)$ as the query.

To tackle this task, knowledge graph embedding models are developed by learning representations of entities and relations within an embedding space. Given an atomic formula $r(s, o)$ from the knowledge graph with $r \in \mathcal{R}$ and $s, o \in \mathcal{E}$, we denote $e_s, e_o, e_r \in \mathbb{R}^d$ as the embedding vectors corresponding to the entities and relation. The estimated embedding of $(s, r, ?)$ is first computed by $f_t(e_s, e_r)$, where $f(\cdot, \cdot) : \mathbb{R}^d \times \mathbb{R}^d \to \mathbb{R}^d$ is the transformation function. Then, the likelihood of $r(s, o)$ is computed by the scoring function $f_s(f_t(e_r, e_s), e_o)$, where $f_s(\cdot, \cdot) : \mathbb{R}^d \times \mathbb{R}^d \to [-\infty, +\infty]$ is the scoring function related to the embedding space.

## 2.5 NEURAL SYMBOLIC SEARCH WITH FUZZY LOGIC

The t-norm $\top : [0, 1] \times [0, 1] \to [0, 1]$ is the continuous relaxation of logical conjunction $\wedge$, which aims to generalize classical two-valued logic by allowing intermediary truth values between 1 (truth) and 0 (falsity). A common example of a T-norm is the product-norm, defined as $a\top_P b = a * b$. Following the fuzzy logic, the negation can be relaxed as $1 - x : [0, 1] \to [0, 1]$. The T-conorm is the dual to t-norm for disjunction and is defined by $\bot(x, y) : 1 - (1 - x)\top(1 - y)$. Details of t-norm can refer to Appendix G. Then we can define the truth value function $T$ as the following:

**Definition 6** (Truth value function). *Let $\phi$ and $\psi$ be existential formulas, $\top$ and $\bot$ are t-norms and t-conorms, $\bot^\star$ is another t-conorm, and $r \in \mathcal{R}$, $a, b \in \mathcal{E}$, with $P_r(a, b)$ representing the truth value of $r(a, b)$. The truth value function $T$, whose range is $[0, 1]$, is defined as follows:*

*(i)* $T(r(a, b)) = P_r(a, b)$
*(ii)* $T(\neg\phi) = 1 - T(\phi)$
*(iii)* $T(\phi \wedge \psi) = T(\phi)\top T(\psi)$, $T(\phi \vee \psi) = T(\phi)\bot T(\psi)$
*(iv)* $T(\exists x\phi(x)) = \bot^\star_{a \in \mathcal{E}} T(\phi(a))$

Combining fuzzy logic operations with the KG embedding models and using the max operation as another $t$-conorm $\bot^\star$ (Yin et al., 2024), verifying whether $s$ is the answer to a query involving many existential variables can be viewed as a multi-variable optimization problem:

$$T(q(s)) = \max_{x_i \in \mathcal{E}, 1 \leq i \leq n} T(c_1)\top \cdots \top T(c_k). \tag{3}$$

The maximum truth value of the answer $a \in \mathcal{A}[\phi(y)]$ should be 1. Conversely, the maximum truth value of an incorrect candidate entity should be 0.

## 3 NEURAL LOGICAL INDICES REDUCES THE SEARCH DOMAIN

We define the search domain $\mathcal{D}$ as the set of candidate entities when the symbolic search algorithm operates over the variables. We denote $\mathcal{D}_x$ and $\mathcal{D}_y$ as the search domain for $x$ and $y$, respectively. We argue treating the entire entity set $\mathcal{E}$, as done by previous search algorithms (Bai et al., 2023; Yin et al., 2024), is unnecessary because each variable must maintain consistency with its surrounding constraints (Chen et al., 2011). Overall, for each variable, our framework extracts the surrounding subgraph as constraints and computes the neural logical indices by these constraints to reduce the search domain.

## 3.1 Neural Logical indices

Given a query with the variable set $V = \{x_1, \cdots, x_n, y\}$ over KG, neural logical indices are defined as a mapping from a variable in $V$ to a subset of the entity set $\mathcal{E}$: $\mathcal{I}(x) : V \to 2^{\mathcal{E}}$. To compute the $\mathcal{I}(x)$ for each variable, we conceptually define the constraints as the subgraph pattern $G_x^S$ from the query graph $G_\phi$. Then we propose the strategy to apply neural embedding models $h$ to identify the entities that satisfy the constraints represented by $h(G_x^S)$. Let $h(G_x^S))$ denote the ranking of the entities, we determine $\mathcal{I}(x)$ by selecting the top $k$ entities from this ranking: $\mathcal{I}(x) = \textbf{Topk}(h(G_x^S), k)$. Thus, $\mathcal{I}(x)$ serves as our reduced search domain $\mathcal{D}_x$, which can accelerate symbolic search algorithms.

By using neural logical indices as the search domain for variables, we can narrow down the search space, thereby simplifying the optimization problem in Equation 3 as follows:

$$T(\phi_i(c/y; x_1, \cdots, x_n)) = \max \quad T(c_1)\top \cdots \top T(c_k) \tag{4}$$
$$\text{s.t.} \quad x_i \in \mathcal{D}_i, 1 \le i \le n, \tag{5}$$

where $c \in \mathcal{D}_y$ and the domain $\mathcal{D}x_i$ is simply denoted by $\mathcal{D}_i$.

The realizations of neural logical indices involve flexible choices for selecting subgraphs as constraints and methods for computing them. We introduce two strategies: local and global. An example in Fig. 2 illustrates these two types of constraints. We define the computation of neural logical indices as the CUTDOMAIN function, which can be easily integrated with symbolic search algorithms.

## 3.2 Local constraints strategy: relation tail prediction task

The local constraints strategy only includes the first-order neighbor of the variable, where $G_y^S = \mathcal{N}_n(x_i, G_\phi)$, as illustrated in Fig. 2. We only consider the information from relations. It is evident that since $y$ must be the tail of the "Graduate" relation and the head of the "Married" relation, searching the variable $y$ within these entities is equivalent to searching the entire space $\mathcal{E}$.

To utilize the relations to prune the search domain, we propose the relation tail prediction task which predicates the tail only given the relation. [2] To address the incompleteness of this task, we adopt the knowledge graph embedding framework, as they share similar characteristics. Instead of starting from scratch, we train a hyper-network (Ha et al., 2016) to generate new embeddings $h$ based on old KG embedding models (Trouillon et al., 2016; Chen et al., 2021). Given the entity embedding $e_o$ and relation embedding $e_r$ in pre-trained knowledge graph embeddings, we have

$$\hat{\mathbf{e}}_o = \text{RELU}(\mathbf{W_1}\mathbf{e}_o + \mathbf{b_1}), \hat{\mathbf{e}}_r = \text{RELU}(\mathbf{W_2}\mathbf{e}_r + \mathbf{b_2}).$$

Then the likelihood of relation tail pair $f_s(\hat{e}_r, \hat{e}_t)$ is computed by the same scoring function in Section 2.4. For a possible entity $e_o$, we employ the T-norm to calculate the scores as follows: $\sigma(f_s(\text{Mar}\hat{\text{ried}}, \hat{e}_o)\top\sigma(f_s(\text{Gra}\hat{\text{duate}}, \hat{e}_o))$, wherer $\sigma(\cdot)$ is the sigmoid activate function.

## 3.3 Global constraints strategy: query embedding

The global constraints strategy extends the constraints to encompass the entire query graph. As illustrated in Fig. 2, the utilized constraints represent the whole graph, which means that $G_y^S = G_\phi$ for $y$. Although the problem formulation has become more complex, we can leverage the ability of query embedding $h$ to directly address. Following this, the two-stage coarse-to-fine ranking process is implemented, similar to the coarse-to-fine ranking used in information retrieval (Liu et al., 2019).

## 4 Search with Neural Logic Index

In this section, we introudce an efficient framework called Neural Logical Index for Search Approximately (NLISA). We begin by reviewing relevant results from FIT (Yin et al., 2024). Next, we explain how to accelerate the existing steps using neural logical indices. Following this, we discuss the appropriate approach for searching cyclic queries in sub-problem optimization. Finally, we present the complexity analysis and discuss the differences compared to existing approximate methods.

---

[2] Predicting the head of relation $r_+$ can be modeled as the tail of reverse relation $r_-$.

## 4.1 RECAP OF NEURAL SYMBOLIC SEARCH METHOD

**Definition 7** (Leaf node). *A leaf node is a variable node connecting to only one other variable node.*

**Definition 8** (Fuzzy vector). *Given the domain $\mathcal{D}$ and a membership function $\mu : \mathcal{D} \to [0, 1]$, we represent the fuzzy set of $\mathcal{D}$ as vector form $D$ with $D_i = \mu(s, \mathcal{D})$, where $s \in \mathcal{D}$.*

The key technique of FIT (Yin et al., 2024) is that constant nodes and leaf nodes can be removed, with the corresponding constraints stored in fuzzy vectors. The complexity of removing constant nodes and leaf nodes are $O(|\mathcal{E}|)$ and $O(|\mathcal{E}|^2)$, respectively. By continuously removing constant and leaf nodes, FIT can handle acyclic queries. For cyclic queries, FIT enumerates one variable within the cycle as a constant node, which results in exponential complexity.

## 4.2 SUB-PROBLEM REDUCTION WITH NEURAL LOGICAL INDICES

We observe that the truth values of the atomic formula in previous works (Bai et al., 2023; Yin et al., 2024) can be interpreted as normalization using the average, as shown following. Denoting observed KG as $G_{\text{train}}$ and the observed tail set as $T_s^r = \{t|(s, r, t) \in G_{\text{train}}\}$, the truth value $P_r(s, o) \in [0, 1]$ can be obtained from the normalization:

$$P_r(s, o) = \frac{exp(f_s(f_t(e_r, e_s), e_o))}{\sum_{o_i \in T_s^r} exp(f_s(f_t(e_r, e_s), e_{o_i}))/|T_s^r|}. \tag{6}$$

The normalization can be adjusted as needed; further details are provided in Appendix D.

Then we can solve the problem in Equation 4 by brute force, which involves the $n + 1$ order tensor $\mathbf{T}^\phi$, where $\mathbf{T}^\phi_{i_1, i_2, \cdots, i_n, i_{n+1}} = T(\phi_i(y^{i_{n+1}}; x_1^{i_1}, \cdots, x_n^{i_n}))$ and $x_1^{i_1}$ denote the $i_1$ entity in $\mathcal{D}_i$. Though the search domain is reduced, the complexity is also huge. We sequentially optimize the problem for given $x_i$ and the sub-problem can be extracted by the following:

$$\max_{x_i \in \mathcal{D}_i} [T(c_1) \top \cdots \top T(c_k)] = \max_{x_i \in \mathcal{D}_i} [\top_{c_i \in \mathcal{N}(x_i)} T(c_i)] \top [\top_{c_j \notin \mathcal{N}(x_i)} T(c_j)]. \tag{7}$$

Denote the number of totally involved variables in this sub-problem as $k$, the involved tensor is $k$ order. When $x_i$ is a leaf variable, only the second-order tensor is computed to solve the sub-problem and the second tensor is necessary when computing atoms with two variables as arguments. Then eliminated constraints are stored in the fuzzy vectors.

**Proposition 1** (REMOVELEAFNODE). *The leaf node in $G_\phi$ can be removed in $O(|\mathcal{D}_x|^2 + |\mathcal{D}_x||\mathcal{D}_y|)$.*

It's clear that our realization for removing nodes has **lower complexity** when $|\mathcal{D}| < |\mathcal{E}|$. The details of implementation for REMOVELEAFNODE function can refer to Appendix C. And we also have similar results REMOVECONSTNODE for constant node (Yin et al., 2024).

**Proposition 2** (REMOVECONSTNODE). *The constant node in $G_\phi$ can be removed in $O(|\mathcal{D}_x| + |\mathcal{D}_y|)$.*

## 4.3 APPROXIMATE SEARCH FOR CYCLIC QUERIES

The cyclic queries can not be addressed by removing constant nodes and leaf nodes (Yin et al., 2024), and it encounters exponential complexity. To tackle this, we propose local search over local constraints and autoregressively search the assignment for variables. With the $x_i$ as an example, we optimize the most likely entity for every $o \in \mathcal{D}_y$ over the remaining constraints and fuzzy vectors as follows:

$$\max_{x_i} [\top_{c_i \in \mathcal{N}_e(x_i)} T(c_i)] \top [\top_{x \in \mathcal{N}_n(x_i)} \mu(x, C_x)]. \tag{8}$$

Specifically, we treat the other existential nodes $\{x_j\}$ in $\mathcal{N}_n(x_i)$ as dummy nodes, respectively applying the max operation to $\{x_j\}$ to eliminate the variables. Then we take max operation over the target variable $x_i$ to determine its assignment. This approach effectively exploits information from local constraints and breaks the cycle while maintaining quadratic complexity with respect to the search domain. Once all remaining variable assignments are obtained, we use t-norm and t-conorm to integrate the truth values of the logical query under the given variable assignments.

Realization of the above approximate search induces a function LOCALOPTIMIZE.

---

**Algorithm 1** Neural logical Index enhanced Search Approximately (NLISA)

---

**Require:** Input query graph $G_\phi$ and initial fuzzy vectors for existential variables and free variable as $C_x$ and $C_y$, the size of the reduced domain $|\mathcal{D}_x|$ and $|\mathcal{D}_y|$.
**Ensure:** Output answer vector $T(G_\phi, \{C_x\}, C_y)$
    $(\{\mathcal{D}_x\}, \mathcal{D}_y) \leftarrow \text{CUTDOMAIN}(G_\phi, |\mathcal{D}_x|, |\mathcal{D}_y|)$
    $(G_\phi, \{C_x\}, , C_y) \leftarrow \text{REMOVECONSTNODE}(G_\phi, \{C_x\}, C_y)$
    **while** $G_\phi$ contains a leaf node $x_i$ **do**
        $(G_\phi, \{C_x\}, C_y) \leftarrow \text{REMOVELEAFNODE}(x_i, G_\phi, \{C_x\}, C_y)$
    **end while**
    **for** each remaining node $x_j$ in $G_\phi$ **do**
        $(G_\phi, \{C_x\}, C_y) = \text{LOCALOPTIMIZE}(x_j, G_\phi, \{C_x\}, C_y)$
    **end for**
    **return** $T((G_\phi, \{C_x\}, C_y))$

---

## 4.4 ALGORITHM AND COMPLEXITY ANALYSIS

Finally, we present the complete procedure of our method, as shown in Algorithm 1. Our objective is to propose an efficient symbolic search method for **general EFO1 queries**. The key aspect is that we can flexibly **reduces the search domain**. Additionally, our appropriate search reduces the complexity of answering cyclic queries from exponential to **quadratic with respect to the search domain**.

The space complexity of our method is $\mathcal{O}((|\mathcal{E}| + |\mathcal{R}| + d_h) * d)$, where $d$ is the embedding dimension of the KG embedding model and $d_h$ is the hidden dimension of hyper-network. This complexity is the same as the knowledge graph embedding and scales linearly with the sizes of entities and relations.

The time complexity is given by $\mathcal{O}((|\mathcal{D}_x||\mathcal{D}_y| + |\mathcal{E}_x|^2 + |\mathcal{D}_x| + |\mathcal{D}_y|)d)$, where $|\mathcal{D}_x|$ and $|\mathcal{D}_y|$ are the size of search domain of the existential and free variable, respectively. It is evident that our algorithm exhibits quadratic complexity for any EFO1 query and can flexibly reduce the complexity by adjusting the size of the search domain.

Compared to other approximate symbolic search methods, such as CQD-beam (Arakelyan et al., 2020) and CQD-CO (Arakelyan et al., 2020), CQD-beam is constrained by the operator tree and only utilizes partial constraints to retain consistent entities as intermediate variables. On the other hand, CQD-CO optimizes the representation of existential variables in continuous space, which may limit expressiveness and result in slower performance. Notably, both methods do not reduce the search domain; instead, they only decrease the amount of information retained for intermediate variables.

## 5 EXPERIMENTS SETTING

In this section, we conduct a comprehensive evaluation of our method across diverse tasks to investigate its effectiveness and efficiency. In terms of query structure, we consider tree-form queries and general EFO1 queries. Regarding the scale of the knowledge graph, we consider graphs with 15,000, 60,000, and 400,000 entities.

## 5.1 BENCHMARKS

The BetaE benchmark (Ren & Leskovec, 2020) is the standard benchmark for complex question answering (CQA), primarily containing tree-form queries. The benchmark is comprised of three knowledge graphs (KGs): FB15k (Bordes et al., 2013), FB15k-237 (Toutanova et al., 2015), and NELL995 (Xiong et al., 2022). Specifically, the BetaE benchmark contains 14 distinct query types, with 5 types involving negation operations. It is important to note that the "pni" query type in BetaE is a universal first-order logic query, and is therefore excluded from the evaluation.

The Real EFO1 benchmark (Yin et al., 2024) proposes 10 new query types beyond the tree-from queries, with the same KGs as BetaE. In particular, the Real EFO1 benchmark introduces new patterns, including multi-graph, and cyclic graphs. The visualization of these query structures of BetaE and Real EFO1 benchmark is presented in Appendix E.

The Smore benchmark (Ren et al., 2022) considers the same tree-form queries as BetaE benchmark, but the queries are sampled from a much larger-scale KG. Since only the FB400K dataset with 40,0000 entities has been released, we select this KG as the large-scale benchmark.

## 5.2 Evaluation Protocol

To evaluate the effectiveness over incomplete knowledge graphs (KGs), we adopt the evaluation scheme from (Ren & Leskovec, 2020), distinguishing the answers to each query into easy and hard sets. Hard queries are defined as non-trivial queries that cannot be answered by direct traversal along the edges of the KG and require predicting at least one missing link in the test and validation splits. We assess the CQA models on these non-trivial queries by calculating the rank $r$ for each hard answer against non-answers, and we compute the Mean Reciprocal Rank (MRR) and HIT@k.

To evaluate the efficiency, we experimentally measure the speed and running memory required to answer complex queries. We sample 100 queries for different query types in each benchmark and calculate the average queries per second (QPS). We record both the model memory and the maximum running memory during evaluations. For a fair comparison between models that support batching and those that do not, the running memory is calculated by excluding the model memory from the total running memory and then dividing the result by the batch size.

## 5.3 Baselines

We consider various state-of-the-art CQA methods as baselines. In particular, we compare our approach with strong baselines from symbolic search methods, including CQD-CO (Arakelyan et al., 2020), CQD-Beam (Arakelyan et al., 2020),QTO (Bai et al., 2023), and FIT (Yin et al., 2023). Additionally, we include ConE and LMPNN as baselines for comparison with query embedding methods. We also examine the GNN-QE method, which combines graph neural networks with symbolic methods. To ensure fairness, we use the same checkpoint for those methods requiring a pre-trained neural link predictor, including CQA, QTO, FIT, and our method. For the FB15K, FB15K-237, and NELL datasets, we utilize the checkpoints provided by CQD-CO (Arakelyan et al., 2020). For FB400K, the details for the pretrained checkpoint are provided in Appendix F. Since FIT is equivalent to QTO in tree-form queries (Yin et al., 2024), we don't distinguish them in this case.

## 6 Results

Our experimental results reveal two key insights. First, the search domain can be significantly reduced with minimal performance loss, thereby alleviating efficiency and scalability issues. Second, in addressing cyclic queries, our proposed approximate search exhibits quadratic complexity with respect to the search domain and achieves performance comparable to that of precise symbolic search methods. To simplify the analysis, we set $|\mathcal{D}_x| = |\mathcal{D}_y|$ as approximately 10% of the corresponding entity size. Specifically, we set $\mathcal{D}_x$ as 2000, 2000, and 6000 for FB15k-237, FB15k, and NELL, respectively. We present the results of three benchmarks in the following three sections. The first insight is demonstrated across all three benchmarks, while the second insight is illustrated on the Real EFO1 benchmark in Section 6.2. In the implementation, both NLISA (Local) and NLISA (Global) utilize the local constraints strategy for existential variables. However, NLISA (Local) applies the local constraints strategy for free variables, while NLISA (Global) employs the global constraints.

## 6.1 Tree Form Queries: BetaE benchmark

We present the results of efficiency and performance on BetaE benchmark in Table 1 and Table 2, respectively. Table 1 shows that symbolic search methods, including CQD-CO and FIF, exhibit remarkably low QSP, while FIT suffers from rapid CUDA memory usage as the size of the knowledge graph (KG) increases. In contrast, our methods, which utilize neural logical indices, significantly reduce the search domain, improving QSP and decreasing CUDA memory usage. Table 2 demonstrates that our method with the global constraint strategy achieves an average performance of 90% of symbolic search method across three KGs, nearly surpassing all baselines except for the precise search methods. Our method with the local constraint strategy does not rely on query embedding methods, achieving lower performance but demonstrating higher QSP compared to those methods.

Table 1: Efficiency results of the Tree-Form queries on BetaE benchmark (Ren & Leskovec, 2020).

| KG | Metric | ConE | LMPNN | CQD-CO | FIT | NLISA(local) | NLISA(global) |
|---|---|---|---|---|---|---|---|
| FB15k-237 | Queries per Second ↑ | 100 | 105 | 14 | 31 | **115** | 86 |
| | CUDA Memory of Running (M) ↓ | 458 | 355 | **176** | 1208 | 232 | 337 |
| FB15K | Queries per Second ↑ | 97 | 101 | 10.1 | 20 | **109** | 79 |
| | CUDA Memory of Running (M) ↓ | 338 | 365 | **189** | 2068 | 255 | 381 |
| NELL | Queries per Second ↑ | 23 | 24 | 7 | 3 | **35** | 20 |
| | CUDA Memory of Running (M) ↓ | 1635 | 1955 | 760 | 15324 | **774** | 953 |

Table 2: MRR results(%) of the Tree-Form queries on BetaE benchmark (Ren & Leskovec, 2020). The scores of the baselines are taken from their papers. We differentiate the methods that have complexity concerning $|\mathcal{E}|$. We highlight the best results in red and the second-best results in blue.

| Method | 1p | 2p | 3p | 2i | 3i | ip | pi | 2u | up | AVG.(P) | 2in | 3in | inp | pin | AVG.(N) |
|---|---|---|---|---|---|---|---|---|---|---|---|---|---|---|---|
| | | | | | | FB15K-237 | | | | | | | | | |
| CQD-CO | 46.7 | 9.6 | 6.2 | 31.2 | 40.6 | 16.0 | 23.6 | 14.5 | 8.2 | 21.9 | | | | | |
| ConE | 41.8 | 12.8 | 11.0 | 32.6 | 47.3 | 25.5 | 14.0 | 14.5 | 10.8 | 23.4 | 5.4 | 8.6 | 7.8 | 4.0 | 5.9 |
| LMPNN | 45.9 | 13.1 | 10.3 | 34.8 | 48.9 | 17.6 | 22.7 | 13.5 | 10.3 | 24.1 | 8.7 | 12.9 | 7.7 | 4.6 | 8.5 |
| CQD-Beam | 46.7 | 13.3 | 7.9 | 34.9 | 48.6 | 20.4 | 27.1 | 17.6 | 11.5 | 25.3 | | | | | |
| GNN-QE | 42.8 | 14.7 | 11.8 | 38.3 | 54.1 | 18.9 | 31.1 | 16.2 | 13.4 | 26.8 | 10.0 | 16.8 | 9.3 | 7.2 | 8.5 |
| FIT/QTO | 46.7 | 14.6 | 12.8 | 37.5 | 51.6 | 21.9 | 30.1 | 18.0 | 13.1 | 27.4 | 14.0 | 20.0 | 10.2 | 9.5 | 13.4 |
| NLISA(Local) | 46.6 | 14.0 | 12.2 | 32.1 | 44.4 | 20.2 | 26.4 | 16.9 | 11.7 | 24.9 | 11.4 | 14.7 | 9.3 | 8.0 | 10.9 |
| NLISA(Global) | 46.6 | 14.3 | 12.3 | 36.2 | 50.0 | 21.2 | 29.3 | 17.6 | 12.5 | 26.7 | 12.4 | 17.1 | 9.3 | 8.3 | 11.8 |
| | | | | | | FB15K | | | | | | | | | |
| CQD-CO | 89.2 | 25.6 | 13.6 | 77.4 | 78.3 | 44.2 | 33.2 | 41.7 | 22.1 | 46.9 | | | | | |
| ConE | 75.3 | 33.8 | 29.2 | 64.4 | 73.7 | 50.9 | 35.7 | 55.7 | 31.4 | 49.8 | 17.9 | 18.7 | 12.5 | 9.8 | 14.8 |
| LMPNN | 85.0 | 39.3 | 28.6 | 68.2 | 76.5 | 43.0 | 46.7 | 36.7 | 31.4 | 50.6 | 29.1 | 29.4 | 14.9 | 10.2 | 20.9 |
| CQD-Beam | 89.2 | 65.3 | 29.7 | 77.1 | 80.6 | 71.6 | 70.6 | 72.3 | 59.4 | 68.5 | | | | | |
| GNN-QE | 88.5 | 69.3 | 58.7 | 79.7 | 83.5 | 70.4 | 69.9 | 74.1 | 61.0 | 72.8 | 44.7 | 41.7 | 42.0 | 30.1 | 39.6 |
| FIT/QTO | 89.4 | 65.6 | 56.9 | 79.1 | 83.5 | 71.8 | 73.1 | 73.9 | 59.0 | 72.5 | 40.2 | 38.9 | 34.8 | 28.1 | 35.5 |
| NLISA(local) | 87.2 | 61.0 | 53.1 | 67.6 | 75.4 | 66.2 | 65.0 | 52.6 | 42.6 | 63.4 | 44.8 | 39.9 | 36.8 | 33.0 | 38.6 |
| NLISA(Global) | 89.4 | 64.7 | 55.4 | 76.4 | 82.3 | 71.1 | 71.1 | 71.9 | 57.5 | 71.1 | 48.3 | 44.4 | 37.9 | 34.5 | 41.3 |
| | | | | | | NELL | | | | | | | | | |
| CQD-CO | 60.4 | 17.8 | 12.8 | 39.3 | 46.6 | 22.1 | 30.1 | 17.3 | 13.2 | 28.8 | | | | | |
| ConE | 53.1 | 16.1 | 13.9 | 40.0 | 50.8 | 26.3 | 17.5 | 15.3 | 11.3 | 27.2 | 5.7 | 8.1 | 10.8 | 3.5 | 6.4 |
| LMPNN | 60.6 | 22.1 | 17.5 | 40.1 | 50.3 | 24.9 | 28.4 | 17.2 | 15.7 | 30.8 | 8.5 | 10.8 | 12.2 | 3.9 | 8.9 |
| CQD-Beam | 60.4 | 22.6 | 13.6 | 43.6 | 53.0 | 25.6 | 31.2 | 19.9 | 16.7 | 31.8 | | | | | |
| GNN-QE | 53.3 | 18.9 | 14.9 | 42.4 | 52.5 | 18.9 | 30.8 | 15.9 | 12.6 | 28.9 | 9.9 | 14.6 | 11.4 | 6.3 | 10.6 |
| FIT/QTO | 60.8 | 23.8 | 21.2 | 44.3 | 54.1 | 26.6 | 31.7 | 20.3 | 17.6 | 33.4 | 12.6 | 16.4 | 15.3 | 8.3 | 13.2 |
| NLISA(Local) | 60.3 | 23.2 | 19.9 | 43.2 | 53.5 | 26.6 | 32.3 | 20.3 | 17.2 | 32.9 | 10.3 | 14.3 | 13.9 | 6.5 | 11.3 |
| NLISA(Global) | 60.8 | 23.3 | 19.8 | 44.0 | 53.8 | 26.8 | 32.6 | 20.3 | 17.3 | 33.2 | 10.2 | 14.2 | 14.0 | 6.5 | 11.3 |

## 6.2 GENERAL EFO1 QUEIRES: REAL EFO1 BENCHMARK

Here, we present our results in Table 3 on the real EFO1 benchmark, which contains general EFO1 queries beyond tree-form queries. For baselines QTO and ConE relied on the operator tree, the new queries cannot be represented by an operator tree and can only be **syntactically approximated**. Table 3 shows that our methods achieve nearly 95% of the performance of symbolic search methods, outperforming all other baselines. This result indicates that our reduction framework is superior to the tree-structure approximation used by QTO. Regarding cyclic queries, including "3c" and "3m", our approximate search framework achieves an average performance of almost 95% compared to FIT, validating our second insight. Our method even outperforms over "2il" and "3il", which we hypothesize is due to the half-precision used in FIT to reduce memory usage. In terms of efficiency, our method demonstrates significantly higher QSP compared to FIT, as shown in Fig. 1.

## 6.3 LARGE SCALE TREE-FORM QUERIES: SMORE BENCHMARK

Since training query embeddings on large-scale knowledge graphs (KGs) is challenging, we can only consider existing query embedding methods and symbolic search using knowledge graph embeddings as baselines. For the FB400K dataset, which contains 400,000 entities, we set $|\mathcal{D}_x| = |\mathcal{D}_y| = 8000$, representing only 2% of the total. Table 4 demonstrates that precise symbolic search methods face out-of-memory issues due to their quadratic complexity with respect to $|\mathcal{E}|$. This can be anticipated based on the rapid increase in CUDA memory usage as the number of entities grows, as shown in Table1.

Table 3: MRR results(%) of the queries on the real EFO1 benchmark (Yin et al., 2024). The scores of the baselines are directly taken from Yin et al. (2024). We differentiate the methods that have complexity concerning $|\mathcal{E}|$. We highlight the best results in red and the second-best results in blue.

| Method | pni | 2il | 3il | 2m | 2nm | 3mp | 3pm | im | 3c | 3cm | AVG. |
|---|---|---|---|---|---|---|---|---|---|---|---|
| FB15K-237 | | | | | | | | | | | |
| CQD-CO | 7.7 | 29.6 | 46.1 | 6.0 | 1.7 | 6.8 | 3.3 | 12.3 | 25.9 | 23.8 | 16.3 |
| ConE | 10.8 | 27.6 | 43.9 | 9.6 | 7.0 | 9.3 | 7.3 | 14.0 | 28.2 | 24.9 | 18.3 |
| LMPNN | 10.7 | 28.7 | 42.1 | 9.4 | 4.2 | 9.8 | 7.2 | 15.4 | 25.3 | 22.2 | 17.5 |
| QTO | 12.1 | 28.9 | 47.9 | 8.5 | 10.7 | 11.4 | 6.5 | 17.9 | 38.3 | 35.4 | 21.8 |
| FIT | **14.9** | **34.2** | **51.4** | **9.9** | **12.7** | **11.9** | **7.7** | **19.6** | **39.4** | **37.3** | **23.9** |
| NLISA(Local) | 12.3 | 34.2 | 51.0 | 9.8 | **10.0** | 10.3 | 7.4 | 18.1 | 34.7 | 34.7 | 22.3 |
| NLISA(Global) | **14.0** | **34.6** | **51.8** | **10.0** | 9.3 | **10.7** | 7.5 | **18.4** | 35.7 | 35.7 | **22.8** |
| FB15K | | | | | | | | | | | |
| CQD-CO | 7.7 | 29.6 | 46.1 | 6.0 | 1.7 | 6.8 | 3.3 | 12.3 | 25.9 | 23.8 | 16.3 |
| ConE | 37.0 | 40.1 | 57.3 | 33.3 | 11.5 | 23.9 | 27.6 | 38.7 | 35.0 | 36.3 | 34.1 |
| LMPNN | 38.7 | 43.2 | 57.8 | 40.3 | 7.9 | 24.0 | 30.5 | 48.4 | 32.2 | 30.9 | 35.4 |
| QTO | 48.2 | 49.5 | 68.2 | 64.6 | 19.4 | 48.5 | 53.7 | 73.9 | 53.3 | 54.9 | 53.4 |
| FIT | **57.9** | **70.4** | **77.6** | **73.5** | **39.1** | **57.3** | **64.0** | **79.4** | **63.8** | **65.4** | **64.8** |
| NLISA(local) | 55.9 | 69.1 | 70.5 | 66.0 | **38.3** | 49.1 | 55.2 | 76.0 | 57.9 | 59.7 | 59.8 |
| NLISA(Global) | **60.7** | **70.4** | **76.1** | **68.8** | 36.2 | **51.0** | **57.0** | **79.9** | **60.3** | **61.5** | **62.2** |
| NELL | | | | | | | | | | | |
| CQD-CO | 7.9 | 48.7 | 68.0 | 31.7 | 1.5 | 12.9 | 13.8 | 33.9 | 38.8 | 35.9 | 29.3 |
| ConE | 10.3 | 42.1 | 65.8 | 32.4 | 7.0 | 12.6 | 16.8 | 34.4 | 40.2 | 38.2 | 30.0 |
| LMPNN | 11.6 | 43.9 | 62.3 | 35.6 | 6.2 | 15.9 | 19.3 | 38.3 | 39.1 | 34.4 | 30.7 |
| QTO | 12.3 | 48.5 | 68.2 | 38.8 | 12.3 | 22.8 | 19.3 | 41.1 | 45.4 | 43.9 | 35.3 |
| FIT | **14.4** | **53.3** | 69.5 | **42.1** | **12.5** | **24.0** | 22.8 | 41.5 | **47.5** | **45.3** | **37.3** |
| NLISA(Local) | 13.9 | 53.1 | **71.6** | 40.6 | **11.9** | **23.1** | **23.0** | 40.6 | 46.0 | 44.1 | 36.8 |
| NLISA(Global) | **14.0** | **53.4** | **72.1** | **41.0** | 10.7 | 22.7 | **23.1** | 40.6 | 46.0 | 44.1 | 36.8 |

Table 4: Averaged MRR results(%) on large KGs with different methods. The results of GQE, Q2B, and BetaE are taken from Ren et al. (2022). The CQD-CO and NLISA(Local) use the same backbone of knowledge graph embedding.

| | GQE | Q2B | BetaE | CQD-CO | FIT/QTO | NLISA(Local) |
|---|---|---|---|---|---|---|
| FB400K | 36.0 | 51.7 | 50.5 | 43.3 | OOM | **58.9** |

Our method, which employs a local constraints strategy, outperforms both the query embedding methods and CQD-CO. This underscores the scalability challenges encountered by precise symbolic search methods and highlights the importance of effective pruning techniques.

## 7 CONCLUSION

Complex query answering over knowledge graphs is a crucial multi-hop reasoning task aimed at addressing first-order logical queries within large and incomplete knowledge graphs. Although symbolic search methods exhibit strong performance and expressiveness, they face efficiency challenge that hinders further development and application. Query embedding models learn representations for complex queries, offering fast speed but often providing only generic performance. In this paper, we integrate their advantage together in a mutually beneficial way and propose an approximate search framework combined with flexible neural logical indices to address these efficiency issues. The neural logical indices can be computed rapidly using embedding methods, significantly reducing the search domain of symbolic methods. In particular, the approximate search framework can handle cyclic queries well with a quadratic complexity. Our approximate search is precise for acyclic queries. Experiments on various benchmarks show that, with a 10% reduced search domain, our method achieves 90% performance, including for cyclic queries, while the QSP assess efficiency is improved to be comparable to that of query embedding methods. Additionally, we demonstrate that our framework can execute neural-symbolic methods on a KG with an order of magnitude more entities than before, highlighting the scalable nature of our approach.

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

# A    RELATED WORK

## A.1    QUERY EMBEDDING METHODS

Answering complex logical queries over knowledge graphs is naturally extended from link prediction and aims to handle queries with complex conditions beyond simple link queries. This task gradually grows by extending the scope of complex logical queries, ranging from conjunctive queries (Hamilton et al., 2018) to Existential Positive First-Order (EPFO) queries (Ren et al., 2020), Existential First-Order (EFO) queries (Ren & Leskovec, 2020), real Existential First-Order queries (Yin et al., 2024). The primary method is query embedding, which maps queries and entities to a low-dimensional space. The form of embedding has been well investigated, such as vectors (Hamilton et al., 2018; Chen et al., 2022; Bai et al., 2022), geometric regions (Ren et al., 2020; Zhang et al., 2021; Liu et al., 2021), and probabilistic distributions (Ren & Leskovec, 2020; Choudhary et al., 2021; Yang et al., 2022; Wang et al., 2023). These methods not only explore knowledge graphs embedding but also leverage neural logical operators to generate the embedding of complex logical queries.

## A.2    SYMBOLIC METHODS

Neural-symbolic CQA models represent variables as fuzzy sets with vector forms $\mu \in [0,1]^{\mathcal{E}}$ and apply fuzzy logic to model the logic operations naturally. Built on the operator tree, GNN-QE (Zhu et al., 2022) represents the intermediate variable as a fuzzy vector, and simultaneously adapts graph neural network from KG completion to execute relation projection and models the logical operations with fuzzy logic. In particular, the neural symbolic search method combining the knowledge graph embedding with symbolic search is of particular interest. CQD-beam (Arakelyan et al., 2020; 2023) uses beam search during the execution of operator tree, maintaining only a beam width of entities for intermediate variables. CQD-CO uses gradient optimization to estimate the embedding of existential variables (Arakelyan et al., 2020). Unlike previous approximate search methods, QTO is a precise symbolic search method because it finds that the tree-form queries can be solved on $O(|\mathcal{E}|^2)$. FIT (Yin et al., 2024) extends its scope to general EFO1 by using the enumeration to handle cyclic queries. However, the complexity is $|\mathcal{E}|^n$ (Yin et al., 2024), where $n$ is the variable number of query. In general, the above two precise methods constantly remove nodes and preserve results with fuzzy vectors corresponding to variables. While symbolic methods demonstrate good performance and strong interpretability, they struggle with high computational complexity.

There are many other models and datasets proposed to enable answering queries with good performance and additional features, see the comprehensive survey Ren et al. (2023).

# B    DETAILS OF CONSTRAINTS STRATEGIES

For local constraints, when the variable involves logical disjunction, we use the T-conorm to handle it.

For global constraints, the computation of existential variables should treat them as free variables, while the original free variables are converted into existential variables. This allows us to use query embedding to compute the global constraints for the existential variables.

# C    DETAILS OF FUNCTIONS

REMOVCONSTNODE($G_\phi, |\mathcal{D}_x|, |\mathcal{D}_y|$) removes the constant nodes based on the given domain of variables. We consider the simplest case where $(s, r, e, \text{NEG} = \text{FALSE})$, with $s$ as the constant node, $r$ as the relation, $e$ as the existential variable, and $\text{NEG} = \text{FALSE}$ indicating that this edge is positive. We first construct the symbolic representation of this edge: $\boldsymbol{s} = [P_r(s, e^i)] \in \mathbb{R}^{|\mathcal{D}_e|}, e^i \in \mathcal{D}_e$. We then update this representation into the fuzzy vector of the existential variable $e$ by t-norm: $\mu(e, \mathcal{D}) = \mu(e, \mathcal{D}) \top \boldsymbol{s}$. Consequently, we can remove this edge while preserving its constraints in the fuzzy vector of $e$. The above method can be generalized to other cases, such as when the variable is a free variable $y$ and the edge is negative. A similar update approach can be used, and specific details can be found in Yin et al. (2024).

REMOVCONSTNODE($G_\phi, |\mathcal{D}_x|, |\mathcal{D}_y|$) removes one leaf node based on the given domain of variables. We consider the most difficult case where ($e_1, r, e_2$, NEG = FALSE), with $s$ as the constant node, $r$ as the relation, $e$ as the existential variable, and NEG = FALSE indicating that this edge is positive. The symbolic representation of this edge results in a matrix: $\boldsymbol{S} = [P_r(e_1^i, e_2^j)] \in \mathbb{R}^{|\mathcal{D}_{e_1}| \times |\mathcal{D}_{e_2}|}, e_1^i \in \mathcal{D}_{e_1}, e_2^i \in \mathcal{D}_{e_2}$. We then update this representation into the fuzzy vector of the existential variable $e$ by t-norm and max operation: $\mu(e, \mathcal{D}) = \mu(e, \mathcal{D}) \top \max(\boldsymbol{S}, axis = 0)$. The proof of the effectiveness of the above update can be found in Yin et al. (2024), and this update can be easily extended to other cases, such as when the variable is a free variable $y$ and the edge is negative.

# D  DETAILS OF TRUTH VALUES

The symbolic representation in the previous symbolic search method QTO and FIT is constructing the set of truth values matrix for the whole knowledge graph To convert real number scores computed by knowledge graph embedding modes to truth value that falls into $[0, 1]$, QTO/FIT use the softmax function: $P_{r,a}^\star(b) = \frac{exp(s(a,r,b))}{\Sigma_{c \in \mathcal{E}} exp(s(a,r,c))}$. Next, QTO and FIT scale the results of the softmax function using a factor based on the observed edges in the training graph since softmax outputs a vector that sums to 1: $\mathcal{G}_o$.

$$Q_{a,b} = \begin{cases} \frac{|\{d|(a,r,d) \in \mathcal{G}_o\}|}{\Sigma_{c \in \{d|(a,r,d) \in \mathcal{G}_o\}} P_{r,a}^\star(c)}, & \text{if } |\{d|(a,r,d) \in \mathcal{G}_o\}| > 0 \\ 1, & \text{if } |\{d|(a,r,d) \in \mathcal{G}_o\}| = 0 \end{cases} \tag{9}$$

where $E_{a,r} = \{b \mid (a,r,b) \in \mathcal{G}_o\}$ represents the set of observed edges. Then the $a$-th row of $r$-th matrix is got by clamping the value for each element:

$$P_r(a, b) = min(1, P_{r,a}^\star(b) \times Q_{a,b}) \tag{10}$$

They then mark the observed edges and set the truth value for these edges to 1. The scaling and marking operations are performed on a case-by-case basis for each fact, which cannot be parallelized.

We demonstrate that these scaling operations can be parallelized through caching. For cases where $|\{d|(a,r,d) \in \mathcal{G}_o\}| > 0$: the truth value is computed by the following normalization:

$$Q_{a,b} = \frac{exp(f_r(a,b))}{\sum_{\{d|(a,r,d) \in \mathcal{G}_o\}} exp(f_r(a,d))}. \tag{11}$$

To simplify the calculations, we use log-scale operations

$$log(Q_{a,b}) = f_r(a,b) - log(\sum_{\{d|(a,r,d) \in \mathcal{G}_o\}} exp(f_r(a,d))). \tag{12}$$

If we cache $S_a^r = log(\sum_{\{d|(a,r,d) \in \mathcal{G}_o\}} exp(f_r(a,d)))$ with $|\mathcal{E}| \times |\mathcal{R}|$ size, we can parallel index the cached values to optimize the computation of Equation 12. For cases where $|\{d|(a,r,d) \in \mathcal{G}_o\}| > 0$, we have:

$$S_a^r = log(\sum_{\{d \in \mathcal{E}\}} exp(f_r(a,d))), \tag{13}$$

which allows the scaling operation to yield the softmax result. In general: the cached values $S \in R^{\mathcal{E} \times |\mathcal{R}}$ are defined as follows:

$$S_a^r = \begin{cases} log(\sum_{\{d|(a,r,d) \in \mathcal{G}_o\}} exp(f_r(a,d))), & \text{if } |\{d|(a,r,d) \in \mathcal{G}_o\}| > 0 \\ log(\sum_{\{d \in \mathcal{E}\}} exp(f_r(a,d))), & \text{if } |\{d|(a,r,d) \in \mathcal{G}_o\}| = 0 \end{cases} \tag{14}$$

This caching mechanism grows linearly with the size of the knowledge graph. By utilizing this caching strategy, we can perform parallel computations for the scaling operations, facilitating efficient dynamic symbolic representation construction. However, when it comes to listing the marked training facts, our symbolic representation may exhibit slightly lower performance compared to the original construction.

We formula the constructing process in the following three steps For the case that $|\{d|(a,r,d) \in \mathcal{G}_o\}| > 0$, the above derivation is equivalent with Equation 6.

However, the mask created based on the training facts cannot be completed in parallel. Therefore, we rely on previous steps by caching. This is a key issue that may lead to a drop in our method's performance in experiments.

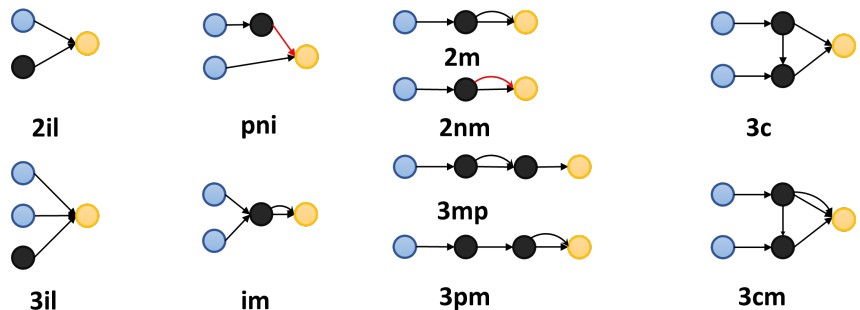

Figure 3: 14 query types propsed in BetaE benchmark (Ren & Leskovec, 2020). These query types are modeled by the operator tree.

Figure 4: 10 query types propsed in Real EFO1 benchmark (Yin et al., 2024). These query types are modeled by query graph.

## E DETAILS OF QUERY STRUCTURES

We present the visualization of query types BetaE benchmark (Ren & Leskovec, 2020) and Real EFO1 benchmark (Yin et al., 2023) in Fig. 3 and Fig. 4, where the visualization of BetaE benchmark is taken from Wang et al. (2023) and the visualization of Real EFO1 benchmark is taken from Yin et al. (2024).

## F DETAILS OF IMPLEMENTATION

### F.1 KNOWLEDGE GRAPH COMPLETION

We reproduce the results from previous work (Chen et al., 2021) to train the embedding and hyper-network. For the FB400K dataset, we search the hyperparameters with the following settings: learning rates of $[1 \times 10^{-1}, 1 \times 10^{-2}]$, embedding dimensions of $[100, 200, 400]$, and $\lambda$ values of $[0.0005, 0.005, 0.01, 0.05, 0.1, 0.5, 1, 0]$. We utilize the ComplEx model with the N3 regularizer. The embedding initialization is set to $1 \times 10^{-3}$, and we employ the Adagrad optimizer.

### F.2 RELATION TAIL PREDICTION

**Why Use Hypernet?**

Relation tail prediction is fundamentally similar to knowledge graph completion (KGC) tasks, as both involve inferring missing tails from a knowledge graph. In KGC, the query is presented as $(h, r, ?)$, to predict the tail $t$. In contrast, relation tail prediction aims to predict the tail $t$ given the relation $r$. This similarity makes the embedding methods used in KGC equally applicable to relation tail prediction. Hypernet is employed to generate weights dynamically, allowing the model to automatically adjust representations for new tasks. This facilitates the sharing and adaptation

Table 5: MRR results for the trained Hypernet.

|       | FB15K-237 | FB15K | NELL | FB400K |
|-------|-----------|-------|------|--------|
| MRR   | 1.0       | 0.998 | 0.99 | 1.0    |

of weights across different tasks. By adopting the embeddings from KGC, Hypernet enhances the utilization of existing knowledge and enables quicker adaptation to relation tail prediction. Additionally, Hypernet requires only a few parameters, improving performance while reducing training costs and storage needs.

### How Is Hypernet Trained?

The training process is similar to that of KGC, where each triple (h,r,t)(h,r,t) is replaced with (r,t)(r,t). This requires only minor modifications compared to the existing KGC framework.

### What Is the Performance of Hypernet?

For the hypernet used for relation tail prediction, we set the learning rate to $1 \times 10^{-2}$ and search the hidden dimensions $[100, 200, 400]$ also using the ComplEx model with the N3 regularizer. The embedding initialization remains at $1 \times 10^{-3}$, and the optimizer is Adagrad. The performance of our method is quite impressive, as demonstrated in Table 5. The results indicate a strong capability of the hypernet in addressing relation tail prediction tasks.

## G   $t$-NORM INTRODUCTION

**Definition 9** ($t$-norm). *A $t$-norm $\top$ is a function: [0,1] x [0,1] $\rightarrow$ [0,1] that satisfies the following properties:*

> *(i) Communitavity: $a\top b = b\top a$*
>
> *(ii) Monotonicity: $(a\top b) \leq (c\top b)$ if $a \leq c$*
>
> *(iii) Associativity: $(a\top b)\top c = a\top(b\top c)$*
>
> *(iv) Neutrality: $a\top 1 = a$*

Then the $t$-conorm $\bot$ is directly defined by $a\bot b = 1 - (1-a)\top(1-b)$, which follows the De Morgan's law.

Finally, we introduce some common $t$-norms which are of interest:

> (i) Godel: $a\top_G b = \min(a, b)$
>
> (ii) Product: $a\top_P b = a * b$
>
> (iii) Łukasiewicz: $a\top_{LK} b = \max(a + b - 1, 0)$

In the main paper, we mainly focus on the Godel and Product $t$-norm.

