# OpenReview forum: "Neural Logical Index for Fast Knowledge Graph Complex Query Answering"
_ICLR.cc/2025/Conference — Submitted to ICLR 2025_

### Official Review · Reviewer_ecuA · 2024-11-03

**Soundness:** 3
**Presentation:** 3
**Contribution:** 2
**Rating:** 5
**Confidence:** 4

**Summary:**

The paper addresses the challenge of efficiently answering complex queries over large and incomplete knowledge graphs (KGs). The authors propose a novel framework that combines the speed of embedding methods with the precision of symbolic search approaches. Their method introduces Neural Logical Indices (NLI) to preemptively reduce the search domain using embedding-based techniques. The framework approximates cyclic queries in quadratic complexity, achieving significant efficiency gains with minimal performance loss. Experiments across multiple benchmarks demonstrate the scalability and performance of their approach.

**Strengths:**

1. The method reduces computation costs by 90%, significantly improving efficiency with minimal performance loss.
2. It is scalable to larger knowledge graphs, handling orders of magnitude more entities than traditional methods.
3. The introduction of Neural Logical Indices effectively narrows the search domain, making symbolic computation more feasible.
4. It achieves a good balance between the precision of symbolic search and the speed of embedding methods for tree-form queries.

**Weaknesses:**

1. Even with the efficiency improvements, the framework incurs some performance loss when handling more complex cyclic queries. This limitation could affect use cases where high accuracy on intricate query patterns is essential.
2. The method’s dependency on embedding-based approaches may limit its adaptability and generalizability across different query types or diverse knowledge graph structures. Variability in embedding quality could impact overall performance and effectiveness.
3. The integration of local and global constraints in the framework adds a layer of implementation complexity. This could present challenges during practical deployment, requiring sophisticated engineering and optimization efforts to ensure seamless functionality.
4. While the framework achieves competitive results, it still falls short of the performance of state-of-the-art models in certain scenarios. This performance gap indicates room for further refinement, particularly in enhancing the handling of complex and cyclic queries to match or surpass the best-performing methods available.

**Questions:**

See the Weaknesses.

---

> ### Author Response · Authors · 2024-11-20
> **First Response to  Reviewer ecuA**
>
> We thank you for your time and review. We believe your suggestions have improved our paper.  However, there seem to be some core misunderstandings about our major claim that we would like to clarify.
>
> The following are the rebuttals to your questions.
>
> >Even with the efficiency improvements, the framework incurs some performance loss when handling more complex cyclic queries. This limitation could affect use cases where high accuracy on intricate query patterns is essential.
>
> Thank you for raising the dicussion of complex cyclic queries. Our contribution, in comparison to FIT, lies in developing efficient algorithms while maintaining comparable performance, `which improves the QPS of cyclic queries by nearly 35-40 times`, as shown in the following table.
>
> FIT should be regarded as the upper bound of our framework since its search algorithms are optimal for the perfect symbolic representation. However, FIT suffers from `significant complexity issues`, particularly with exponential complexity in cyclic queries. As shown in the QPS results for FB15k-237 in Table 1, cyclic queries (3c and 3cm) are nearly 30 times slower than other types, and this issue worsens as the size of the knowledge graph increases, resulting `great scalability trouble`. Thus, our framework was proposed to `address efficiency and scalability`, as demonstrated in the table below, which shows that our framework reduces the QPS of cyclic queries by nearly 35-40 times, making their QPS comparable to that of other query types. To ensure a fair comparison, we utilize the same pre-trained model with identical checkpoints. `Given the same symbolic representation capabilities, it is overly ambitious to expect an approximate solving algorithm to outperform FIT on cyclic queries while achieving a tenfold reduction in the search domain.` Attaining 95% performance is already a satisfactory result.
>
> | Query types     | pni     | 2il      | 3il     | 2m     | 2nm     | 3mp   | 3pm     | im      | 3c      | 3cm      |
> | -------- | -------- | -------- | --------| -------- | -------- | -------- | -------- | -------- | -------- | -------- |
> | FIT  | 36.1  | 89.3  | 79.4  | 58.1 |23.1  | 59.5  | 57.5  | 58.5  | 3.4  | 2.9  |
> | NLISA (Global)   | 217.4  | 243.9  | 232.6  | 158.7 | 149.3 |163.9 | 127.6  | 126.6  | 129.9  | 102.2  |
>
> Table 1: Queries Per Second  on the Real EFO1 Benchmark (FB15K-237)
>
> Overall, regarding cyclic queries, our framework achieves a `35-40 times improvement in QPS` while `maintaining 95% performance` compared to FIT. We believe that the considerations of efficiency and scalability discussed in our paper offer valuable insights for the development of CQA models.
>
>
> > The method’s dependency on embedding-based approaches may limit its adaptability and generalizability across different query types or diverse knowledge graph structures. Variability in embedding quality could impact overall performance and effectiveness.
>
> During the execution of query embedding methods, we will `project unseen query types into tree-form queries` that have been thoroughly trained for query embedding.
>
> When applying query embedding to address unseen query types, our framework first constructs an `approximated operator tree` by removing certain edges from the original query graph to simplify it into a tree-form query. The modified query is then transformed into this approximated operator tree. The original answers are included in the approximated operator tree because they satisfy the simplified constraints. As a result, query embedding models encode the approximated operator tree instead of directly encoding the unseen query types, demonstrating robustness across different query types. You can review the results of the query embedding models for ten unseen query types in Table 3 of the main text.
>
> The query embedding is just used for the `rough rank` and just needs to recall the answers in the search domain where the search domain is not small; thus, the requirements for query embedding methods are minimal. Additionally, the strong performance of NLISA (global) on the real EFO1 datasets, as shown in Table 3, supports these arguments, given that the ten query types are new to the query embedding models used.

---

> > ### Author Response · Authors · 2024-11-20
> > **The remaining part of First Response to Reviewer ecuA**
> >
> > > The integration of local and global constraints in the framework adds a layer of implementation complexity. This could present challenges during practical deployment, requiring sophisticated engineering and optimization efforts to ensure seamless functionality.
> >
> > The local and global constraints are computed using `well-established knowledge graph embedding and query embedding models`. The resulting indices represent a widely used `caching technique` in modern database systems. Our neural logical index serves as a neural enhancement to existing indices. In terms of real-world engineering implementation, the impact of adding neural logical indices has already been modularized, making it straightforward to employ.
> >
> > >While the framework achieves competitive results, it still falls short of the performance of state-of-the-art models in certain scenarios. This performance gap indicates room for further refinement, particularly in enhancing the handling of complex and cyclic queries to match or surpass the best-performing methods available.
> >
> > Our method does have a performance gap compared to FIT because we simplify certain aspects of the algorithms. We speculate that there is an improvement in certain query types because our approach can utilize higher precision float32, while FIT uses float16 for memory efficiency. However, this simplification becomes great for cyclic queries, as we provide an approximate solution specifically tailored for them. It is challenging to enhance performance while simultaneously reducing great computation.

---

> ### Comment · Reviewer_ecuA · 2024-11-26
>
> Thank you for your response. Query Embedding essentially addresses the efficiency of regular graph query searches using vector representations. However, its performance remains poor, generally below 50%. Therefore, improving its performance is more critical than further reducing its efficiency.

---

> > ### Comment · Reviewer_foBD · 2024-11-26
> > **Same**
> >
> > I also have similar concerns. It could be convinced if the authors could show that improving efficiency enables us to scale up the training set and gain performance improvement.

---

> > > ### Author Response · Authors · 2024-11-28
> > > **Thanks for raising your concerns.**
> > >
> > > Our discussed symbolic search algorithms are train-free and do not require a training set to scale up. However, the improved efficiency enables the framework to handle larger KGs, as demonstrated in the table below. When faced with the FB400K dataset, which contains 400,000 entities, the FIT method encounters an out-of-memory (OOM) issue, while our method continues to perform effectively.
> > >
> > > ---
> > >
> > > |        | GQE  | Q2B  | BetaE | CQD-CO | FIT/QTO | NLISA(Local)  |
> > > |--------|------|------|-------|--------|---------|---------------|
> > > | FB400K | 36.0 | 51.7 | 50.5  | 43.3   | OOM     |58.9 |
> > >
> > > Table 2 Averaged MRR results(\%) on large KG with 400,000 entities of different methods.

---

> ### Author Response · Authors · 2024-11-26
>
> Thank you for your valuable feedback!
>
> First, the SOTA symbolic search method, FIT, suffers from significant efficiency issues. Its query complexity for cyclic queries is exponential, while its data complexity is quadratic relative to the number of entities in the knowledge graph. Additionally, FIT `results in very slow processing times for cyclic queries` due to this high complexity and cannot process cyclic queries in parallel. When handling 60,000 entities, `it have to use float16 to reduce memory usage even for a 32GB GPU, which leads to substantial scalability challenges`. Therefore, it is essential to develop efficient symbolic methods to address these efficiency and scalability concerns. Our method achieves a 35-fold improvement in efficiency metrics for cyclic queries on FB15k-237 and can scale to knowledge graphs with 400,000 entities. Our framework supports larger knowledge graphs, and the efficiency improvements for cyclic queries can be further enhanced when handling larger datasets.
>
> Second, the performance of current SOTA symbolic search methods, such as FIT, is not poor. These symbolic search methods `enhance the quality of pre-trained knowledge graph embedding models [1] and calibration strategies [2]`, with QTO [1] and CKGR [2] achieving a 25% improvement over  Avg.(P) on FB15k-237.  We utilize older knowledge graph embedding models solely for fairness in evaluation.  Furthermore, `the current evaluation is undervalued` because the evaluated "hard answers" do not encompass all the “hard answers” in true real-world situation. Research [3] conducted under the open-world assumption demonstrates that the mean reciprocal rank (MRR) metric performs better than traditional evaluations. In this context, the open-world assumption means that unknown triplets are considered to include many missing facts that are not present in the training or test sets, resulting in a more realistic setting.
>
> [1] Bai, Y., Lv, X., Li, J., & Hou, L. (2023, July). Answering complex logical queries on knowledge graphs via query computation tree optimization. In International Conference on Machine Learning (pp. 1472-1491). PMLR.
>
> [2] Xiao C, Cao Y. Complex Logical Query Answering by Calibrating Knowledge Graph Completion Models[J]. arXiv preprint arXiv:2410.07165, 2024.
>
> [3] Yang H, Lin Z, Zhang M. Rethinking knowledge graph evaluation under the open-world assumption[J]. Advances in Neural Information Processing Systems, 2022, 35: 8374-8385.

---

> > ### Comment · Reviewer_ecuA · 2024-11-26
> >
> > Thank you for your response. Your method adopts a balanced approach between efficiency and performance. However, I believe you should further elaborate on the total time comparisons in your paper. For CQA tasks, FIT-like CQA methods already reduce processing time significantly compared to traditional query methods because of the embedding characteristic. Whether it is necessary to further reduce time at the expense of performance remains debatable. Although FIT is state-of-the-art, the accuracy of some queries remains at a relatively low level. I would prefer to see an approach that not only outperforms FIT in terms of effectiveness but also addresses efficiency issues to meet the acceptance standard. Efficiency improvements based on FIT’s performance might limit the workload and innovation level of the paper. The narrative of the paper could benefit from further refinement.

---

> ### Author Response · Authors · 2024-11-26
> **Misunderstanding over FIT**
>
> Thank you for the feedback.
>
> However, there are some misunderstanding regarding of FIT.
>
> >For CQA tasks, FIT-like CQA methods already reduce processing time significantly compared to traditional query methods because of the embedding characteristic.
>
>  FIT is **NOT** an embedding method; rather, it is a symbolic method that does not reduce processing time **but significantly rise processing time** compared to  query embedding.  We list the complexity of  each operation for different models in the below Tab.1, which shows that FIT have much more time complexity than query embedding methods like ConE/BetaE and that's the reason why FIT can perform better than query embedding. Furthermore, the number of operation executions for cyclic queries grows exponentially, which causes significant challenges. Overall, FIT suffers from significant efficiency issues and face OOM trouble in below Tab.2. Our method address the efficiency issues and support the KG with 400,000 entities, which show our scalability.
>
> ---
>
> | Model      | Projection                               | Intersection             | Negation            | union               | answer retrieval             |
> |------------|----------------------------|---------------------|-------------------|---------------------|---------------------|
> | ConE/BetaE | O(ld²)                             | O(ild²)            | O(d)              | DNF                 | O(Ed)            |
> | CQD-Beam   | O(Ed)                           | O(Ekd)           | -                | O(Ekd)           | O(Ed)            |
> | GNN-QE     | O(Ed² + Rd)                  | O(E)             | O(E)            | O(E)             | O(E)             |
> | QTO/FIT    | O(E^2)                      | O(E)             | O(E)            | O(E)             | O(E)             |
>
> Table 1 Time complexity of each set operation and answer retrieval. We denote  **E and R as the number of entities and relations**,  the embedding dimension and hidden dimension of MLPs as $d$, number of layers of MLPs/GNNs as $l$, number of branches in an intersection operation as $i$,  the beam size as $k$, number of branches in a union operation as $u$.
>
> ---
>
> |        | GQE  | Q2B  | BetaE | CQD-CO | FIT/QTO | NLISA(Local)  |
> |--------|------|------|-------|--------|---------|---------------|
> | FB400K | 36.0 | 51.7 | 50.5  | 43.3   | OOM     |58.9 |
>
> Table 2 Averaged MRR results(\%) on large KG with 400,000 entities of different methods.

---

### Official Review · Reviewer_BmbY · 2024-11-04

**Soundness:** 3
**Presentation:** 2
**Contribution:** 3
**Rating:** 6
**Confidence:** 3

**Summary:**

This paper addresses the limitations of current approaches to complex query answering, particularly query embedding and neural symbolic search methods. While query embedding methods offer speed, they often suffer from lower performance. Conversely, neural symbolic search methods achieve strong performance but are computationally demanding.
The proposed approach seeks to combine the strengths of both methods, aiming to provide high performance with efficiency. It does this by leveraging surrounding constraints for each query variable to reduce the search space. In one strategy, termed local constraints, embedding models determine constraints via relation tail prediction, based on the variable's connection to the relation. In another strategy, called global constraints, a broader set of constraints is applied across the entire query to further limit the domain.
The resulting framework, Neural Logical Index for Search Approximately (NLISA), can handle both tree and cyclic queries in a scalable way. Extensive experiments across multiple CQA datasets demonstrate NLISA's effectiveness and efficiency compared to several baseline models.

**Strengths:**

– The problem targeted in this paper (enhance the scalability of CQA methods while obtaining a good performance) is a valid and an important ongoing problem existing in the literature of CQAs.

– the paper is well written in terms of motivation and problem statement.

– The proposed approach can handle several query types including cyclic ones where many previous works did not handle.

**Weaknesses:**

– In general, the local version of the model obtain lower performance than state of the art models. For example, on FB15K, the results are very low (approx 55) comparing to other models which obtain approx 72. Thus the scalability is obtained with the cost of substantial performance loss for local model.

– while the paper is written well in terms of problem statement and motivation, the technical details of the approach requires further and better explanation. Moreover, some acronyms should be defined before the first usage to improve the clarity of the paper, e.g., EFO1.

– Scalability remains a significant challenge in the embedding and complex query answering (CQA) domains. Even studies that claim to improve scalability often use only small datasets for evaluation. Given years of advancements in model development, it is expected that models claiming improved scalability would be trained and tested on large-scale datasets. However, most datasets used in this field are quite small. In this paper, even the largest dataset, used to demonstrate model scalability, contains only 400,000 entities—far fewer than real-world knowledge graphs, which often have over 90 million entities.

**Questions:**

Why GNN-QE is not included in table 1?

---

> ### Author Response · Authors · 2024-11-20
> **First Response to Reviewer BmbY**
>
> Dear reviewer, we really appreciate your thorough and constructive feedback. We believe your suggestions have improved our paper. We first want to address your weakness for improvement and then answer your question.
>
> The following are the rebuttals to your weakness.
>
> >In general, the local version of the model obtain lower performance than state of the art models. For example, on FB15K, the results are very low (approx 55) comparing to other models which obtain approx 72. Thus the scalability is obtained with the
> cost of substantial performance loss for local model.
>
> We apologize for the typos regarding the AVG(P) results on the FB15K dataset of BetaE benchmark. The actual result is `63.4, not 55`, as previously stated in the `first general response`.
> NLISA is a general framework that can flexibly extract the subgraph structure surrounding the variables and estimate their domains. Though NLISA (local) only considers the neighbor, it offers advantages in `efficiency, deployment, and scalability`.
>
> NLISA (local) utilizes relational constraints from first-order neighbor subgraphs, `eliminating the need to train query embedding models`. Both knowledge graph embeddings and hypernetworks required for NLISA (local) can be easily trained using well-established knowledge graph embedding frameworks. This framework only requires relation facts in the knowledge graph for pre-training, making it significantly more efficient than the query embedding framework, which necessitates sampling queries along with positive and negative answers as training data.
> Additionally, NLISA provides an ablation study on the considered query structures, highlighting the importance of subgraph selection.
>
> >while the paper is written well in terms of problem statement and motivation, the technical details of the approach requires further and better explanation. Moreover, some acronyms should be defined before the first usage to improve the clarity of the paper, e.g., EFO1.
>
> Thank you for your kind reminder. We have included the full name of EFO1 at its first mention in line 74, and we will ensure to avoid similar issues in the future. We have rewritten part of Section 3 to clarify the definition and computation of neural logical indices, presenting an example of the local version constraints in Figure 2. Additionally, we provided further details on the neural logical indices used in REMOVECONSTNODE and REMOVELEAFNODE in Appendix C.  Additionally, we have provided further details on the computation of truth values,  and implementation in Appendices D and E. We hope these updates enhance the clarity of our method.

---

> > ### Author Response · Authors · 2024-11-20
> > **The remaining part of First Response to Reviewer BmbY**
> >
> > > Scalability remains a significant challenge in the embedding and complex query answering (CQA) domains. Even studies that claim to improve scalability often use only small datasets for evaluation. Given years of advancements in model development, it is expected that models claiming improved scalability would be trained and tested on large-scale datasets. However, most datasets used in this field are quite small. In this paper, even the largest dataset, used to demonstrate model scalability, contains only 400,000 entities—far fewer than real-world knowledge graphs, which often have over 90 million entities
> >
> > Thank you for your constructive feedback.  We also aim to scale knowledge graph  sizes to much larger sizes, approaching those of real-world knowledge graphs. The reason we are using FB400K with 400,000 entities, is that it is  `the largest dataset currently available with query instances` to the best of our knowledge. Datasets such as ogbl-wikikg2, which contains 2,500,604 entities, and Freebase, with 86,054,151 entities, among large-scale CQA benchmarks like SMORE, involve dynamic sampling. This poses challenges for evaluation due to issues with our reproduction.
> >
> > Experimentally, we are confident that we can scale to ogbl-wikikg2, which has 2,500,604 entities.  Both the knowledge graph embedding and hypernetworks required for NLISA (local) can be easily trained using established knowledge graph embedding frameworks. We have successfully trained the KG embedding and hypernet to complete the NLISA (local). The main challenge is the lack of available benchmark data for ogbl-wikikg2, so we plan to sample some queries ourselves.
> >
> > In theory, the space complexity is O(d*(|E|+ |R|+d_h))  and the time complexity is O((|D_x||D_y|+|D_x|^2+|D_x|+|D_y|)* d *N_q), where |E| is the number of entities, |R| is the number of relations, |D_x| is the number of domain for existential variable, |D_x| is the number of domains for free variables, d is the embedding dimension, d_h is the dimension of the hypernet, and N_q is the number of edges in the query graph. The space complexity is `linear with respect to the size of the knowledge graph`, rather than quadratic as seen in FIT/QTO. Although the time complexity is `quadratic in relation to the size of the KG`, there is an adaptive factor to control the search domain to further reduce the complexity. Both `time and space complexities demonstrate good scalability as the size of the KG increases`.
> >
> > The following are the respnse to your questions.
> > > Why GNN-QE is not included in table 1?
> >
> > We were unable to reproduce the GNN-QE results due to device limitations, which prevented us from including the efficiency results in Table 1. We apologize for this oversight.

---

> > > ### Author Response · Authors · 2024-11-29
> > >
> > > We appreciate your valuable feedback and would like to know if our responses and revised submission have addressed your concerns. Your insights are crucial to us, and we are eager to resolve any remaining issues. If you need any additional information, please let us know.

---

### Official Review · Reviewer_p8gq · 2024-11-04

**Soundness:** 3
**Presentation:** 2
**Contribution:** 2
**Rating:** 5
**Confidence:** 3

**Summary:**

This paper introduces a new knowledge graph complex query answering method called NLISA, featured by computationally fast. It is a neural-symbolic framework with local and global constraints. Experiments on the benchmarks show that the NLISA framework could reduce computation by 90% with a minimal performance loss.

**Strengths:**

1. This paper studies an important problem of fast knowledge graph complex query answering, which could benefit the development of the knowledge graph complex query answering methods.
2. The proposed fast knowledge graph complex query answering method is efficient that could reduce the computation by 90%.

**Weaknesses:**

1. The main method is hard to understand. After reading, I find it is not clear how the neural logical indices are achieved.
2. Some method and experiment details are missing(refer to question 3 and 4 ).
3. Though the NLISA is efficient but the complex query answering performances of NLISA are suboptimal. Table 1 shows the NLISA(local) is more efficient that achieves the best efficiency results. But most of the task performance of NLISA(local) in Table 2 are worse than FIT method. Especially the AVG(P) results on FB15k datasets is significantly worse.

**Questions:**

1. In Figure2, should the query graph represent the query “Fine someone who is married to a person who graduated from the same institution”? Since the in the triples (x1, Graduate, y) and (x1, Graduate, x2), the head entity refers to the same variable x1.
2. The $\top$ in Equation (3) and (4) is hard to understand. Does the $\top$ represents operations between the truth values?  How should we interpret the Equation (3) and (4)?
3. What are the entity embedding and relation embedding from equation in line 244 from? is it pre-trained by some methods?
4. How the hyper-network are train? What data are used for the training?

---

> ### Author Response · Authors · 2024-11-20
> **First Response to  Reviewer p8gq**
>
> We thank you for your time and review. However, there seem to be some core misunderstandings about our presentation that we would like to clarify.
>
> The following are the rebuttals to your weakness.
>
> >The main method is hard to understand. After reading, I find it is not clear how the neural logical indices are achieved.
>
> We apologize for any misunderstandings regarding the main methods. We have rewritten part of `Section 3` to clarify the definition and computation of neural logical indices, presenting an example of the local version constraints in Figure 2. Additionally, we provided further details on the neural logical indices used in REMOVECONSTNODE and REMOVELEAFNODE functions in Appendix C. We hope these updates enhance the clarity of our method.
>
> >Some method and experiment details are missing(refer to question 3 and 4 ).
>
> We provide our response to the below rebuttal to questions 3 and 4.
>
> >Though the NLISA is efficient but the complex query answering performances of NLISA are suboptimal. Table 1 shows the NLISA(local) is more efficient that achieves the best efficiency results. But most of the task performance of NLISA(local) in Table 2 are worse than FIT method. Especially the AVG(P) results on FB15k datasets is significantly worse.
>
> We apologize for the typos regarding the AVG(P) results on the FB15K dataset. The actual result is `63.4, not 55`, as previously stated in the `first general response`.
>
>
> NLISA is a general framework that can flexibly extract the subgraph structure surrounding the variables and estimate their domains. Though NLISA (local) only considers the neighbor, it offers advantages in `efficiency, deployment, and scalability`.
>
> NLISA (local) utilizes relational constraints from first-order neighbor subgraphs, `eliminating the need to train query embedding models`. Both knowledge graph embeddings and hypernetworks required for NLISA (local) can be easily trained
> using well-established knowledge graph embedding frameworks. This framework only requires relation facts in the knowledge graph for pre-training, making it significantly more efficient than the query embedding framework, which necessitates sampling queries along with positive and negative answers as training data.
> Additionally, NLISA provides an ablation study on the considered query structures, highlighting the importance of subgraph selection.
>
> The following are the rebuttals to your questions.
>
> >In Figure2, should the query graph represent the query “Fine someone who is married to a person who graduated from the same institution”? Since the in the triples (x1, Graduate, y) and (x1, Graduate, x2), the head entity refers to the same variable x1.
>
> The triple (x1, Graduate, y) has the negation, thus the query is  “Fine someone who is married to a person who graduated from the same institution”? We updated Fig 2 to change the Graudate  with not Graudate in the figures.  We hope this change can clear the presentation of examples.
>
> >The ⊤ in Equation (3) and (4) is hard to understand. Does the ⊤represents operations between the truth values? How should we interpret the Equation (3) and (4)?
>
>   Yes, ⊤ represents fuzzy logical operations between truth values, with a detailed definition provided in Line 186 and  Appendix G. Equation (3) rewrites the logical query from Boolean variables to truth values, allowing logical queries to be addressed using knowledge graph embeddings rather than searching over a discrete knowledge graph. By leveraging knowledge graph embeddings to compute symbolic representations, it becomes possible to find difficult answers. Equation (4) presents the cut domain version of Equation (3).
>
> >What are the entity embedding and relation embedding from equation in line 244 from? is it pre-trained by some methods?
>
> We feel sorry for this misunderstanding. The entity embedding and relation embedding from the equation in line 244 are reused from the embedding of query embedding models, where the resume refers to the introduction in hypernet.  We have updated the statement in Lines 246-248.
>
> >How the hyper-network are train? What data are used for the training?
>
> Thank you for your question, we present the details in a `first general response to all reviews` and also present the details in  Appendix E.

---

> > ### Author Response · Authors · 2024-11-29
> >
> > We greatly appreciate your valuable feedback and would like to know if our responses and revised submission have satisfactorily addressed your concerns. Your insights are essential to us, and we are eager to resolve any remaining issues. If you require any additional information, please don’t hesitate to reach out.

---

### Official Review · Reviewer_foBD · 2024-11-04

**Soundness:** 4
**Presentation:** 4
**Contribution:** 4
**Rating:** 8
**Confidence:** 3

**Summary:**

This paper proposes a novel framework called Neural Logical Index for Search Approximately (NLISA) to tackle the problem of complex query answering over large and incomplete knowledge graphs. The framework leverages the speed of embedding-based methods and the precision of symbolic search methods by reducing the search domain using neural logical indices computed via embedding techniques. This allows for an efficient and scalable approximate search mechanism, particularly for cyclic queries which are traditionally NP-complete. Experiments on various benchmarks demonstrate that the proposed method achieves significant computation reduction with minimal performance loss.

**Strengths:**

1. The combination of embedding-based and symbolic search methods in a novel framework that reduces the search domain using neural logical indices is a creative approach to the existing challenges in complex query answering.
2. The method is well-designed with details on how the neural logical indices are computed using local and global constraints strategies. The framework is theoretically grounded and empirically evaluated on multiple benchmarks.
3. The paper is well-structured and clearly explains the motivation, approach, and results. Definitions, figures, and tables are used effectively to aid comprehension.
4. The proposed method addresses the critical issues of efficiency and scalability in symbolic search methods, demonstrating its potential for real-world applications involving large knowledge graphs.

**Weaknesses:**

This work is solid. Maybe the authors could discuss more about the caching issue mentioned in appendix line 797, which is suggested to be presented in the main body of the paper if the issue really exists.

**Questions:**

Section 3.2 Why to choose hypernetwork? How sensitive are the results to the choice of embedding model used for computing the neural logical indices? Would different embedding techniques lead to significant variations in performance? The embedding model seems to be related to the caching issue.

---

> ### Author Response · Authors · 2024-11-20
> **First Response to Reviewer foBD**
>
> Dear reviewer, we would like to express our gratitude for your time and valuable feedback. Below is our point-by-point response to your questions.
>
> >Maybe the authors could discuss more about the caching issue mentioned in appendix line 797, which is suggested to be presented in the main body of the paper if the issue really exists.
>
> Regarding your concerns about the caching issue, we complemented more details in Appendix D, where introduce how caching  is used and why caching works. Since the page limit of the main body of the paper, we present the details in the Appendix and provide a reference to the main body.
>
> >Section 3.2 Why to choose hypernetwork?
>
> Regarding your questions about hypernet, please refer to the `first general response` to all reviews and we also present it in Appendix E.
>
> >How sensitive are the results to the choice of embedding model used for computing the neural logical indices?
>
> The neural logical indices are not sensitive to the choice of embedding model because the domain size is sufficiently large. We will include an ablation study on the choice of embedding models in the revised paper.
>
> >Would different embedding techniques lead to significant variations in performance?
>
> In our approach, the local version uses knowledge graph embeddings to compute the neural logical indices, while the global version adapts the query embeddings. You can refer to the comparison between the local and global versions for insights into performance variations. The results indicate that the global version achieves better overall performance, while the local version demonstrates better efficacy. This demonstrates that our framework can benefit from the development of embedding models.
>
> >The embedding model seems to be related to the caching issue.
>
> The caching issue arises from the construction of the symbolic representation. We have provided further details in Appendix D.

---

> > ### Comment · Reviewer_foBD · 2024-11-24
> > **Thanks**
> >
> > Thanks for addressing my concerns. I would keep my rating as it is high enough.

---

> > > ### Author Response · Authors · 2024-11-24
> > > **Thanks**
> > >
> > > Thanks for your kind feedback. We appreciate the opportunity to address your concerns.

---

### Author Response · Authors · 2024-11-20
**First General Response to All Reviews**

We would like to thank the reviewers for their insightful comments. We believe your suggestions have improved our paper. In the following, we present the changes made to the revision which are marked in Brown. We have also clarified some misunderstandings regarding common questions.
1. We corrected `the typos of the AVG(P) result for NLISA (local) in FB15K`, as discussed below.
2. We rewrote part of Section 3 to `clarify the definition and computation of neural logical indices`.
3. We updated Fig. 2 to enhance the clarity of our examples.
4. We provide `details about the hypernet in Appendix E`, including its rationale for usage and training process.
5. We updated the details of the functions in the search algorithms in Appendix C, which helps the reader learn the algorithm.
6. We revised the details regarding the computation of truth values in the symbolic representation construction in Appendix D.


Additionally, we identified some misunderstandings that we would like to clarify regarding common questions. We address  `the typos in Table 2 of the main text ` that many reviewers have pointed out. We also provide  `details about the hypernet `, including its usage rationale and training process.


## Typos in Table 2 of the main text

We apologize for the typos in reporting the AVG(P) result for NLISA (local) on positive queries in FB15K, which mistakenly included the average of all query types, including negation queries. The correct result is `63.4, not 55`, and we have verified that there are no other similar mistakes.  In fact, only this AVG(P) was incorrect, while the results for the other query types remain unchanged. Additionally, our code is available in the abstract, allowing for the results to be reproduced. We hope this correction addresses your concerns regarding NLISA (local).

## Why Use Hypernet?
Relation tail prediction is fundamentally similar to knowledge graph completion (KGC) tasks, as both involve inferring missing tails from a knowledge graph. In KGC, the query is presented in the form of (h, r, ?), to predict the tail t. In contrast, relation tail prediction aims to predict the tail t given the relation r. This similarity makes the embedding methods used in KGC equally applicable to relation tail prediction.

Hypernet[1] is employed to dynamically generate weights, allowing the model to automatically adjust representations for new tasks. This facilitates the sharing and adaptation of weights across different tasks. By adopting the embeddings [2] from KGC, Hypernet enhances the utilization of existing knowledge and enables quicker adaptation to relation tail prediction. Additionally, Hypernet requires only a few parameters, improving performance while reducing training costs and storage needs.

## How Is Hypernet Trained?

The training process is similar to that of KGC, with the exception that each triple (h,r,t) is replaced with (r,t). This requires only minor modifications compared to the existing KGC framework[3].

[1] Théo Trouillon, Johannes Welbl, Sebastian Riedel, Éric Gaussier, and Guillaume Bouchard. Complex embeddings for simple link prediction. In International conference on machine learning, pp.2071–2080. PMLR, 2016.

[2] Yihong Chen, Pasquale Minervini, Sebastian Riedel, and Pontus Stenetorp. Relation prediction as an auxiliary training objective for improving multi-relational graph representations. In 3rd Conference on Automated Knowledge Base Construction, 2021. URL https://openreview.net/forum?id=Qa3uS3H7-Le.

[3] David Ha, Andrew M. Dai, and Quoc V. Le. Hypernetworks. ArXiv, abs/1609.09106, 2016. URLhttps://api.semanticscholar.org/CorpusID:208981547.

---

> ### Author Response · Authors · 2024-11-28
> **Second General Response to All Reviews**
>
> Recent discussion raise the concerns regarding to the efficiency trouble and we want to classify its existence and importance.
>
> We first present the complexity of each operation for different methods in Table 1. The table demonstrates that the state-of-the-art symbolic search method, FIT, requires significantly more computation than embedding methods like ConE and BetaE. Furthermore, **FIT suffers from exponential complexity $O(E^q)$ for cyclic queries**, where $E$ is the entities number and $q$ is the size of query graph.
>
> We then illustrate the efficiency challenges with detailed examples. Table 2 shows that **FIT processes cyclic queries 30 times slower than other query types**, while our method reduces the processing time for cyclic queries by a factor of 35. Additionally, Table 3 highlights **FIT's struggles with out-of-memory issues**, whereas our method performs well with a knowledge graph containing 400,000 entities.
>
>
> ---
> | Model      | Projection                               | Intersection             | Negation            | union               | answer retrieval             |
> |------------|----------------------------|---------------------|-------------------|---------------------|---------------------|
> | ConE/BetaE | O(ld²)                             | O(ild²)            | O(d)              | DNF                 | O(Ed)            |
> | CQD-Beam   | O(Ed)                           | O(Ekd)           | -                | O(Ekd)           | O(Ed)            |
> | GNN-QE     | O(Ed² + Rd)                  | O(E)             | O(E)            | O(E)             | O(E)             |
> | QTO/FIT    | O(E^2)                      | O(E)             | O(E)            | O(E)             | O(E)             |
>
> Table 1 Time complexity of each set operation and answer retrieval. We denote  **E and R as the number of entities and relations**,  the embedding dimension and hidden dimension of MLPs as $d$, number of layers of MLPs/GNNs as $l$, number of branches in an intersection operation as $i$,  the beam size as $k$, number of branches in a union operation as $u$.
>
> ---
>
> |        | GQE  | Q2B  | BetaE | CQD-CO | FIT/QTO | NLISA(Local)  |
> |--------|------|------|-------|--------|---------|---------------|
> | FB400K | 36.0 | 51.7 | 50.5  | 43.3   | OOM     |58.9 |
>
> Table 2 Averaged MRR results(\%) on large KG with 400,000 entities of different methods.
>
> ---
>
> | Query types     | pni     | 2il      | 3il     | 2m     | 2nm     | 3mp   | 3pm     | im      | 3c      | 3cm      |
> | -------- | -------- | -------- | --------| -------- | -------- | -------- | -------- | -------- | -------- | -------- |
> | FIT  | 36.1  | 89.3  | 79.4  | 58.1 |23.1  | 59.5  | 57.5  | 58.5  | 3.4  | 2.9  |
> | NLISA (Global)   | 217.4  | 243.9  | 232.6  | 158.7 | 149.3 |163.9 | 127.6  | 126.6  | 129.9  | 102.2  |
>
> Table 3 Averaged QPS results on FB15k-237 with 14,000 entities.

---

### Meta-Review · Area_Chair_Gtv9 · 2024-12-21

**Metareview:**

This paper proposes a framework called Neural Logical Index for Search Approximately (NLISA) to tackle the problem of complex query answering over large and incomplete knowledge graphs. It utilizes embedding methods to compute Neural Logical Indices (NLI) to reduce the search domain for each variable in advance, followed by an approximate symbolic search for fine ranking.

While the reviewers see some merits in the work (e.g., an important problem and the computational efficiency of the proposed method), some critical issues should be addressed. The initial submission contained too many typos, errors, and unclear explanations, which led to substantial updates during the discussion phase. Clarifying the details seems to require several more rounds. Also, the authors should include larger datasets than those used in the current manuscript. Finally, performance is being sacrificed by improving efficiency. There should be a better balance between efficiency and performance.

**Additional Comments On Reviewer Discussion:**

Reviewer foBD and Reviewer ecuA explicitly acknowledged the authors' rebuttal and participated in the discussion. Reviewer BmbY raised an important issue about the size of the datasets used in the experiments. Reviewer foBD and Reviewer ecuA both expressed concerns about the performance of the proposed method in that performance is too much sacrificed to improve efficiency.

---

### Decision · Program_Chairs · 2025-01-22

Reject